# Changes-of-mind in the absence of post-decision evidence

**Nadim A. A. Atiya** [1,‡], **Arkady Zgonnikov** [2,‡], **Denis O'Hora** [2]*, **Martin Schoemann** [3,4],
**Stefan Scherbaum** [3], **KongFatt Wong-Lin** [1]*

**1** Intelligent Systems Research Centre, School of Computing, Engineering and Intelligent Systems, Ulster
University, Magee Campus, Derry ~ Londonderry, United Kingdom, **2** School of Psychology, National
University of Ireland Galway, Galway, Ireland, **3** Department of Psychology, Technische Universität Dresden,
Dresden, Germany, **4** Department of Management/MAPP, Aarhus University, Aarhus, Denmark

☋ These authors contributed equally to this work.
‡ These authors share first authorship on this work.
* denis.ohora@nuigalway.ie (DO); k.wong-lin@ulster.ac.uk (KFW-L)

pcbi.1007149

UNITED STATES

**Data Availability Statement:** All code and data can
be accessed from our Open Science Framework
project: https://osf.io/y385t/.

## Abstract

Decisions are occasionally accompanied by changes-of-mind. While considered a hallmark
of cognitive flexibility, the mechanisms underlying changes-of-mind remain elusive. Previ-
ous studies on perceptual decision making have focused on changes-of-mind that are pri-
marily driven by the accumulation of additional noisy sensory evidence after the initial
decision. In a motion discrimination task, we demonstrate that changes-of-mind can occur
even in the absence of additional evidence after the initial decision. Unlike previous studies
of changes-of-mind, the majority of changes-of-mind in our experiment occurred in trials
with prolonged initial response times. This suggests a distinct mechanism underlying such
changes. Using a neural circuit model of decision uncertainty and change-of-mind behav-
iour, we demonstrate that this phenomenon is associated with top-down signals mediated
by an uncertainty-monitoring neural population. Such a mechanism is consistent with
recent neurophysiological evidence showing a link between changes-of-mind and elevated
top-down neural activity. Our model explains the long response times associated with
changes-of-mind through high decision uncertainty levels in such trials, and accounts for the
observed motor response trajectories. Overall, our work provides a computational frame-
work that explains changes-of-mind in the absence of new post-decision evidence.

## Author summary

We used limited availability of sensory evidence during a standard motion discrimination
task, and demonstrated that changes-of-mind could occur long after sensory information
was no longer available. Unlike previous studies, our experiment further indicated that
changes-of-mind were strongly linked to slow response time. We used a reduced version
of a previously developed neural computational model of decision uncertainty and
changes-of-mind to account for these experimental observations. Importantly, our model
showed that the replication of these experimental results required a strong link between

**Funding:** N.A. was supported by Ulster University Vice-Chancellor's Research Scholarship Award. A. Z. was supported by Government of Ireland Postdoctoral Fellowship GOIPD/2015/481. M.S. and S.S. were supported by the German Research Council DFG grant SFB 940/2. K.W.-L. was supported by the National University of Ireland Galway Moore Institute Research Visiting Fellowship Award, the Northern Ireland Functional Brain Mapping Project 1303/101154803 funded by InvestNI and the Ulster University, The Royal Society International Exchange Scheme under grant IE151293, and COST Action Open Multiscale Systems Medicine (OpenMultiMed) supported by COST (European Cooperation in Science and Technology). The funders had no role in study design, data collection and analysis, decision to publish, or preparation of the manuscript.

**Competing interests:** The authors have declared that no competing interests exist.

changes-of-mind and high decision uncertainty (i.e. low decision confidence), supporting the notion that changes-of-mind are related to decision uncertainty or confidence.

## Introduction

Perceptual decision making is ubiquitous in our daily lives. As our decisions vary in difficulty, so does our ability to make accurate and well-timed responses [1–3]. In some situations, an initial decision can be revised as new evidence arrives [4–11]. For instance, our interpretation of a road sign can change as we approach it, which may result in a change-of-mind. Previous work investigating changes-of-mind has predominantly focused on revising a decision in response to new sensory evidence [7, 9, 10, 12], demonstrating that the frequency of changes-of-mind increases with task difficulty, and that the majority of the changes improve choice accuracy. Such reversals of decisions in response to new evidence have also been previously linked to error correction [4, 5, 13, 14], indicating a strong association between the mechanisms underlying these behaviours.

Signatures of change-of-mind behaviour have previously been explained by cognitive models [7, 9], which extended the drift-diffusion model of decision making [15–17]. A central feature of these models is the temporal accumulation of noisy momentary evidence over time: when the accumulated evidence reaches a prescribed threshold, a choice is made. However, in extended drift-diffusion models [7, 9], evidence accumulation continues after the initial decision, which may lead to a change-of-mind if the accumulated evidence reaches a second prescribed threshold. Therefore, late-arriving sensory information, or any explicitly provided additional evidence [6, 10], can feed into this extended accumulation process, thereby potentially reversing the initial decision. In many situations, however, additional post-decision evidence is not available, and it is not well understood whether models implying post-decision evidence accumulation in changes-of-mind extend to these situations. Thus, the mechanism underlying changes-of-mind in the absence of additional post-decision evidence remains unclear [7, 18].

Recently, the neural correlates of change-of-mind behaviour in humans and primates have been gradually revealed [8, 10]. In particular, fMRI recordings indicated a strong correlation between changes-of-mind and increased activity in the prefrontal cortex [10]. This correlation supported the argument that top-down signals could play an important role in error correction mechanisms [19]. However, the neural mechanism by which these top-down signals lead to a change-of-mind remains elusive. Furthermore, it is still unclear how this mechanism is linked to other metacognitive processes partially mediated by the frontal cortex, particularly, the encoding of decision uncertainty (or confidence) [20–22].

In this work, we investigated changes-of-mind in the absence of additional post-decision sensory evidence. In contrast to previous studies, the majority of changes-of-mind we observed were associated with prolonged initial response times. Using a neural circuit model, we demonstrate that these changes-of-mind can be attributed to neural feedback control mediated by decision uncertainty. This suggests that top-down uncertainty monitoring could play an important role in inducing changes-of-mind in the absence of additional post-decision evidence, which is consistent with recent neurophysiological evidence [10, 19]. Overall, our work provides a computational framework that explains changes-of-mind in the absence of additional post-decision evidence, from the sensory integration stage up to the motor output.

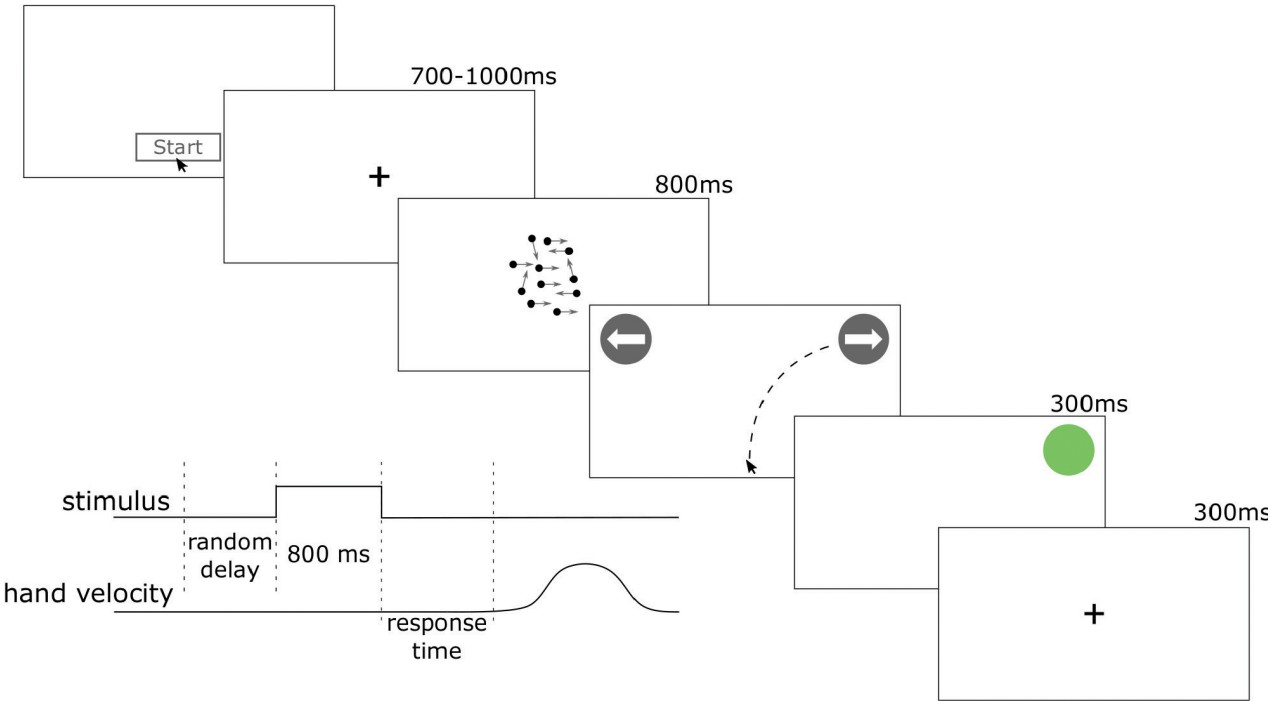

**Fig 1. Experimental setup.** Participants initiated a trial by clicking a start button at the bottom of the screen. After a short random delay (uniformly distributed over 700–1000ms), a random dot kinematogram appeared for 800ms. Participants then chose between two targets: Left or Right. Immediately after the choice, the feedback (red or green circle) was displayed for 300ms, followed by the fixation point (300ms).

## Results

### Experimental results

Eleven participants completed a perceptual decision-making task (2400 trials) in which, upon initiating a trial, a random dot kinematogram (RDK) stimulus [23, 24] appeared after a random delay (uniformly distributed over 700–1000ms). The RDK stimulus was displayed for 800ms, followed by a choice prompt. Participants then decided whether the majority of the dots were moving towards the right or the left. The participants were instructed to respond as quickly and accurately as possible by moving a computer mouse cursor to one of the two choice target locations in the top corners of the computer screen and clicking on it (Fig 1). The difficulty of the task was varied via the *motion coherence* parameter, which controlled the probability of each dot moving in a target direction (left or right); see Methods and materials for details of the experimental setup.

To check the validity of our paradigm, we analysed response times (z-scored within participants) as a function of accuracy and coherence level (Table 1). In our fixed-duration task, the

**Table 1. Parameters of a linear mixed-effects model analysing response time (z-scored within participants) as a function of coherence and choice accuracy.** The model included a random intercept for participant and random slopes for coherence within participant.

|  | Estimate | Std. Error | df | t value | Pr(>|t|) |
|---|---|---|---|---|---|
| Intercept | 0.1737 | 0.0648 | 10.0320 | 2.6813 | 0.0230 |
| Coherence | -0.2302 | 0.1754 | 10.2769 | -1.3124 | 0.2179 |
| Is correct | -0.1232 | 0.0061 | 26260.2576 | -20.0404 | <0.0001 |
| Coherence by Is correct | -0.6067 | 0.0457 | 26260.5434 | -13.2891 | <0.0001 |

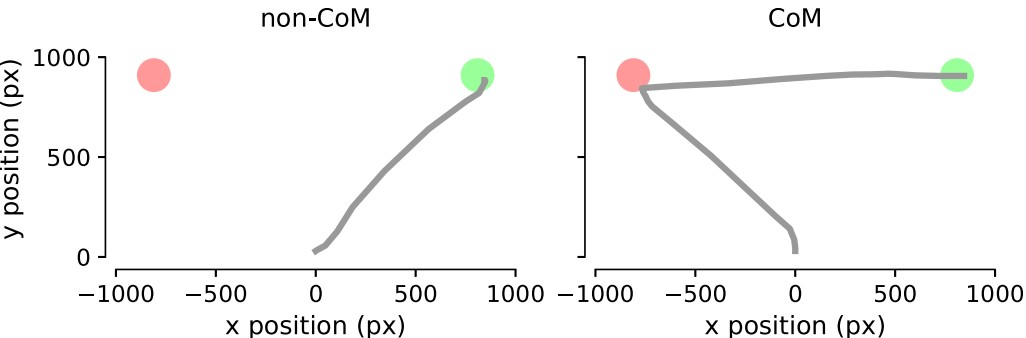

**Fig 2. Response trajectories from two representative trials.** Participants reached one of the two choice targets (red/green areas) directly in non-change-of-mind trials (left), or after reaching towards the opposite target first in change-of-mind trials (right).

term "response time" refers to the time it took participants to initiate a movement towards one of the choice targets after stimulus offset (Fig 1). For correct choices, initial response times were on average faster than in error choices ($b = -0.12$, $t = -20.0$, $p < 0.0001$). There was no evidence for the main effect of coherence ($b = -0.23$, $t = -1.3$, $p = 0.22$); however, we found significant interaction between coherence and choice accuracy ($b = -0.6$, $t = -13.3$, $p < 0.0001$). These results indicate that response time decreased with coherence, but only in correct trials, which is consistent with previous perceptual discrimination studies using the reaction-time task [7, 23].

**Changes-of-mind occur even in the absence of post-decision sensory evidence.** In the vast majority of the trials, participants responded by moving the mouse cursor directly to a choice target. However, in a fraction of trials, participants reversed their initial decision before clicking on one of the choice targets (Fig 2). Similar to previous studies [7, 9, 12, 25], we observed changes-of-mind in 3% trials (0.3% to 6.1%, median 2.4% across 11 participants). However, previous studies investigated changes-of-mind in situations where late-arriving evidence (i.e. due to processing delays) could prompt a reconsideration of the initial decision. Under such circumstances, it was suggested that changes-of-mind can occur within 450ms from initiating the response [7, 26], which strongly links such changes to signal transduction delays. In order to limit the potential effect of late-arriving evidence on changes-of-mind, in our experimental task, participants were instructed to respond only after stimulus offset. Although in some trials the participants initiated their response before stimulus offset, this behaviour was not associated with changes-of-mind. Specifically, participants responded after the stimulus offset in 82% of the trials involving a change-of-mind (as opposed to 70% in trials without a change-of-mind). Importantly, in 72% of these change-of-mind trials, decision reversal occurred later than 450ms after the stimulus offset. Taken together, these observations suggest that in our experiment, the majority of changes-of-mind are not driven by late-arriving post-decision sensory evidence.

In order to examine the relationship between choice accuracy and changes-of-mind, we analysed choice accuracy as a function of coherence level in the presence and absence of changes-of-mind (Table 2, Fig 3a). We found that accuracy increased with coherence ($b = 6.3$, $z = 7.1$, $p < 0.0001$), consistent with previous work on perceptual decision making [23, 24]. In change-of-mind trials, choice accuracy was on average lower than in non-change-of-mind trials ($b = -1.3$, $z = -11.9$, $p < 0.0001$). Moreover, the effect of coherence on accuracy was stronger in non-change-of-mind trials ($b = -6.8$, $z = -8.4$, $p < 0.0001$) than in change-of-mind trials. Despite the less accurate responses in change-of-mind trials, these changes-of-mind were beneficial to the overall performance at intermediate-to-high coherence levels; the accuracy of

**Table 2. Parameters of a generalized linear mixed-effects model analysing choice accuracy as a function of coherence and presence or absence of a change-of-mind.**
The model included a random intercept for participant and random slopes for coherence within participant.

|  | Estimate | Std. Error | z value | Pr(>|z|) |
|---|---|---|---|---|
| Intercept | 1.1751 | 0.1645 | 7.1440 | <0.0001 |
| Is CoM | -1.3436 | 0.1129 | -11.9051 | <0.0001 |
| Coherence | 6.2918 | 0.8918 | 7.0555 | <0.0001 |
| Is CoM by Coherence | -6.7564 | 0.8058 | -8.3852 | <0.0001 |

changes-of-mind was above chance level for 0.128 ($p = 0.005$) and 0.256 ($p = 0.0001$) coherence (see also Fig 3b), demonstrating that changes-of-mind corrected an impending erroneous choice more often than introducing an error. Consequently, in change-of-mind trials, error-to-correct changes were more frequent than correct-to-error changes at 0.128 and 0.256 coherence levels (Fig 3b).

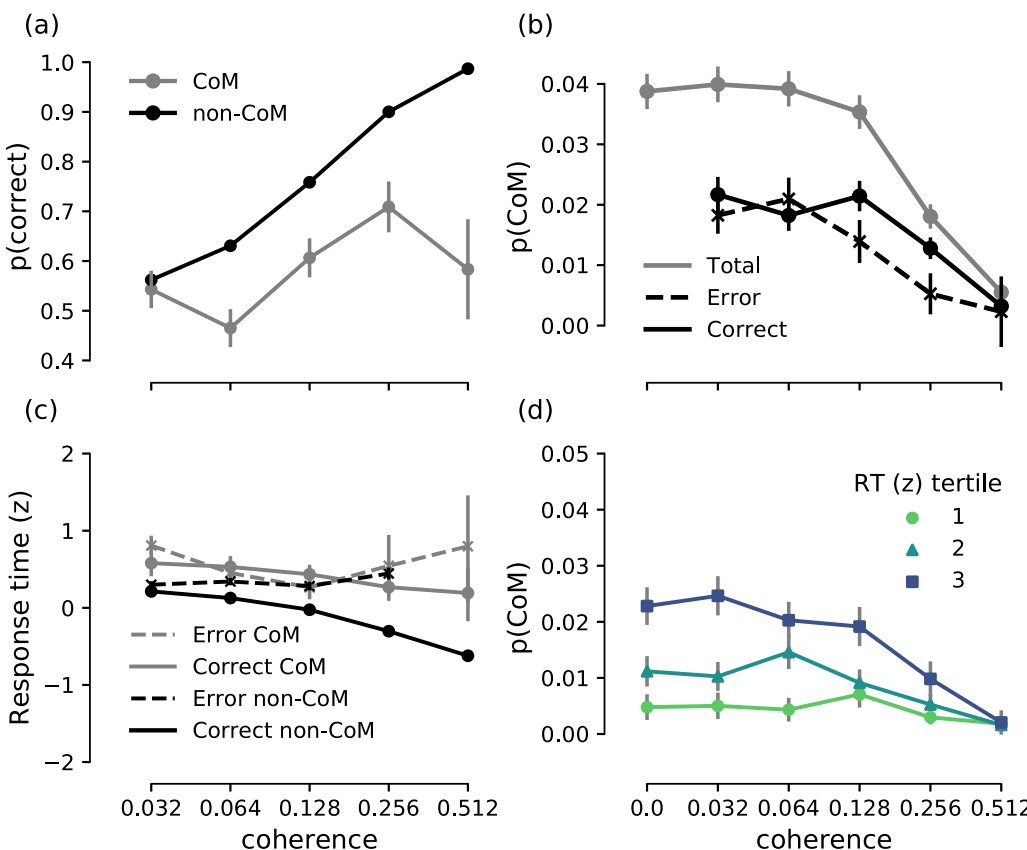

**Fig 3. Experimental results. (a)** Psychometric function showing choice accuracy as a function of coherence level in the presence (grey) and absence (black) of a change-of-mind. Accuracy generally increases as a function of coherence level, but is lower in change-of-mind trials (grey) compared to non-change-of-mind trials (black). However, for intermediate-to-high coherence levels (0.128 and 0.256), the accuracy of change-of-mind trials is above chance level (i.e. >0.5). **(b)** Observed probability of a change-of-mind in all/correct/error trials as a function of coherence. Probability of a change-of-mind peaks at low-to-intermediate coherence levels, and decreases sharply at high coherence levels, with correct changes being more frequent than error changes at moderate to high coherence levels (0.128 and 0.256). **(c)** Response times (z-scored within each participant) for correct and error change-of-mind and non-change-of-mind trials. The '<' pattern of response times in the case of non-change-of-mind trials was consistent with previous observations [7, 23]. **(d)** Observed probability of a change-of-mind as a function of coherence level grouped by the tertile of the initial response time. The majority of change-of-mind trials occurred when response times were longest. In all panels, error bars indicate standard error of mean.

**Table 3. Parameters of a generalized linear mixed-effects model analysing probability of a change-of-mind as a function of coherence and response time (z-scored within participants).** The model included a random intercept for participant and random slopes for response time and coherence within participant.

|  | Estimate | Std. Error | z value | Pr(>|z|) |
|---|---|---|---|---|
| Intercept | -4.0716 | 0.2973 | -13.6968 | <0.0001 |
| RT (z) | 0.4186 | 0.0914 | 4.5799 | <0.0001 |
| Coherence | -2.9697 | 0.5568 | -5.3335 | <0.0001 |

**Changes-of-mind are associated with slow initial decisions.** To clarify the relationship between the formation of an initial decision (as reflected in response times) and the subsequent emergence of a change-of-mind, we analysed the probability of a change-of-mind as a function of coherence level and initial response time (Table 3, see also Fig 3c and 3d). The frequency of changes-of-mind decreased with coherence ($b = -3.0$, $z = -5.3$, $p < 0.0001$). Crucially, changes-of-mind were more likely to occur in trials with a prolonged response time ($b = 0.4$, $z = 4.6$, $p < 0.0001$, see also Fig 3d). This is in stark contrast with previous studies that linked changes-of-mind to fast initial responses [7, 27]. This discrepancy could be attributed to the differences between the experimental tasks as discussed above, and therefore the potentially different mechanisms underlying changes-of-mind. In the next section, we show that our neural circuit model provides a qualitative account of the behavioural results, and provides predictions on the mechanism underlying changes-of-mind in our experimental task.

## Neural circuit model

To shed light on the potential mechanism underlying the observed changes-of-mind, we adopted a previous computational model of decision uncertainty and changes-of-mind [28] (Fig 4). This cortical circuit model was previously shown to account for decision uncertainty and change-of-mind behaviour reported in previous work [7, 11, 22, 29], while capturing recent neurophysiological evidence of encoding decision confidence [10, 19, 22].

In our model, we describe the activity of sensorimotor populations using the two-variable (reduced spiking neural network) model of decision making [30], with two mutually-inhibiting populations selective for leftward/rightward sensory evidence, endowed with self-excitation (Fig 4, cyan box). The hand module consists of two neural populations that receive input from a corresponding left-/right-selective sensorimotor population. Similar to the previous work [28], the hand populations are modelled using a firing-rate type model (i.e. threshold-linear, see Methods). This minimizes the number of new model parameters introduced. The simulated response of the hand populations ultimately determines the model behaviour—in a given simulated trial, a response is recorded when the activity of one of the populations of the hand module reaches a prescribed threshold (see Methods). In our model, we have mapped the output of the neural activity of the motor populations (Fig 4, cyan box) onto the horizontal ($x$) position alongside the mouse cursor trajectories from our experimental data (see Methods). We found that the model could produce motor response trajectories (along the horizontal line) that are qualitatively similar to the experimental ones (Fig 5).

Here, we propose a version of the model which simplifies the neural circuit architecture developed in [28] (see Methods and materials). Specifically, we used only one neuronal population to encode decision uncertainty, termed the *uncertainty-monitoring population* (Fig 4, red circle). We simulated this model (Fig 6) and we observed a phasic neural activity profile for the uncertainty-monitoring population (Fig 6, red activity traces) that is reminiscent of neural recordings from regions or neurons that encode decision uncertainty [19, 22]. In our model, 600ms after stimulus onset, this uncertainty-monitoring population receives and

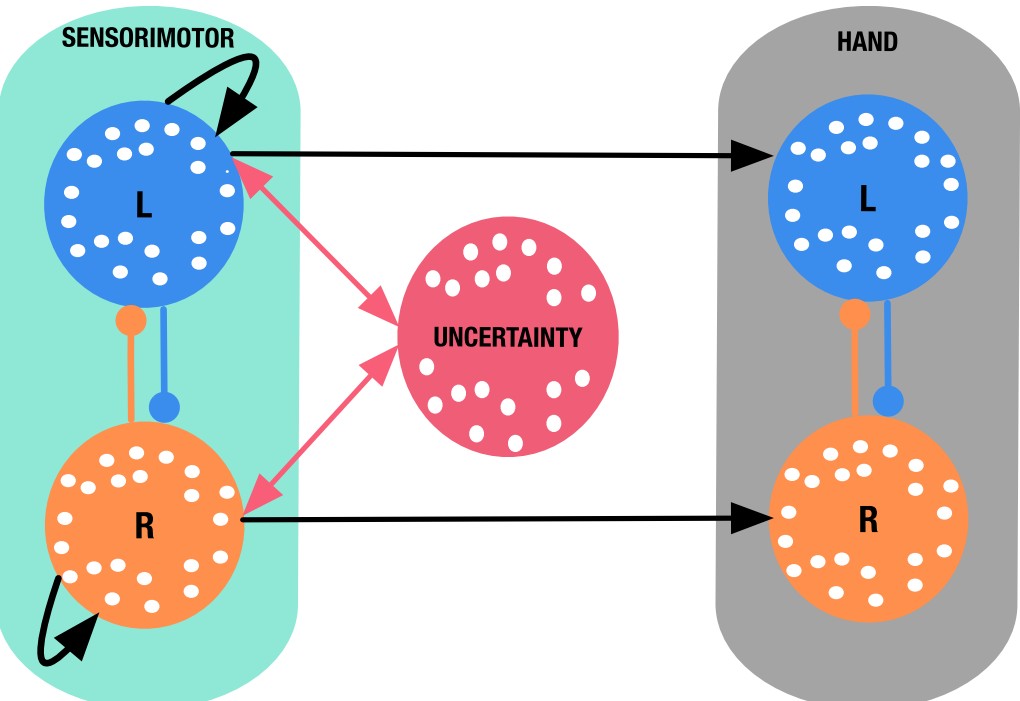

**Fig 4. Simplified neural circuit model of decision uncertainty.** The sensorimotor module (cyan box) consists of two mutually-inhibiting (lines with filled circles) neuronal populations selective for leftward and rightward motion with recurrent excitation (curled black arrows). The uncertainty-monitoring population (red circle) receives summed input from the sensorimotor populations. 600ms after stimulus onset, the summed input is integrated and fed back to the sensorimotor populations (red arrow). The hand response module (grey box) consists of two mutually-inhibiting neuronal populations that integrate the output from the corresponding sensorimotor population. Model results in all subsequent figures were obtained via simulating the model using a single set of parameters (see Table 4 for parameter values).

integrates the summed neural activities of the sensorimotor populations (Fig 4, two-way red arrows), therefore continuously monitoring decision uncertainty during decision making. The uncertainty-monitoring population then in turn equally excites both sensorimotor neuronal populations, effectively providing them with excitatory feedback. When the activity of one of

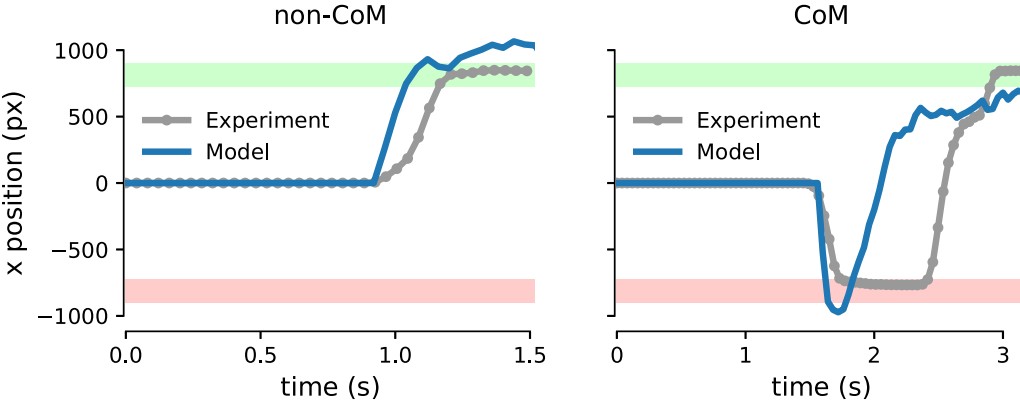

**Fig 5. Experimental mouse cursor trajectories in the x positional space (grey lines with markers) and model-generated motor output (blue solid lines).** See Methods for details on the linear mapping of the firing rates of the model hand response populations onto the x positional space.

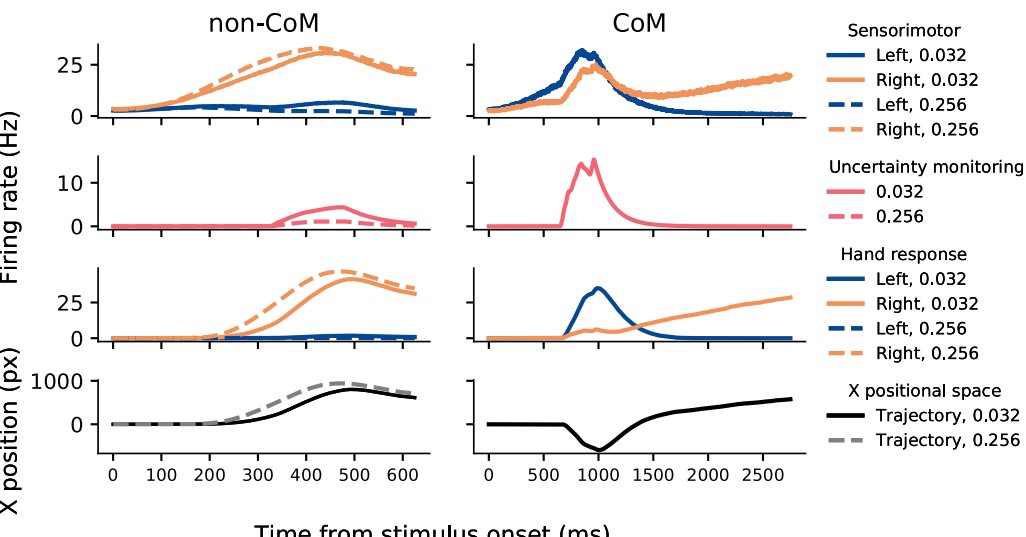

**Fig 6. Neural activity generated by the model.** Activity is averaged over 3168 and 5346 non-change-of-mind trials for 0.032 and 0.256 coherence levels, respectively (left panels), and 120 change-of-mind trials for 0.032 coherence level (right panels). Blue (orange) colours: left (right) neuronal population. In trials without a change-of-mind, the sensorimotor and hand neuronal populations representing the correct (rightward) choice ramp up faster and reach higher activations in the case of high (0.256) coherence level compared to trials with a low (0.032) coherence level. The activity level of the uncertainty-monitoring neuronal population however is greater in trials with low (0.032) coherence. In change-of-mind trials, high uncertainty levels lead to high competition between the left and right sensorimotor neuronal population (through equal feedback excitation). Right panel: Left neuronal population is initially "winning", with a reversal occurring late in the trial. In downstream neuronal populations (for motor output), the left neuronal population reaches choice target, but is eventually suppressed by the rising activity of the right neuronal population. Bottom panel: Model-generated trajectories in the x positional space (see Materials and methods).

the sensorimotor populations reaches a prescribed threshold (42.5 Hz), the uncertainty-monitoring population stops integrating the summed input from the sensorimotor populations. As in the previous modelling work [28], we observed this equal excitation to be maximal in trials with error choices, during difficult tasks, and change-of-mind trials. This is due to the increased absolute input to the uncertainty-monitoring populations under such conditions (Fig 6), in which sufficient time is allowed for the uncertainty-monitoring population to integrate the input (see also [28]).

In quantifying the model's uncertainty as a function of coherence level, we found that uncertainty level increases with coherence for error choices, but decreases for correct choices (Fig 7), which is consistent with previous findings [11, 22, 29]. Furthermore, we found that this positive uncertainty feedback loop is strongly associated with initial response times (Pearson's $r = 0.95$), similar to previous experimental work on decision uncertainty [31]. Taken together, these observations on the relationship between uncertainty, task difficulty, and response time can serve as a basis for explaining the underlying mechanism for changes-of-mind observed in our experimental task. However, in what follows, we first show that the model qualitatively accounts for the behavioural effects observed in our experiment.

**Model accounts for the observed change-of-mind behaviour.** Our neural circuit model readily accounts for the observed experimental findings (Fig 8, cf. Fig 3). In particular, the model reproduces the positive relationship between choice accuracy and coherence, which is weakened in change-of-mind trials (Fig 8a). The probability of error/correct changes-of-mind, as well as total proportion of changes-of-mind as a function of coherence are also captured in the model (Fig 8b). Similarly, the model accounts for the relationship between response time

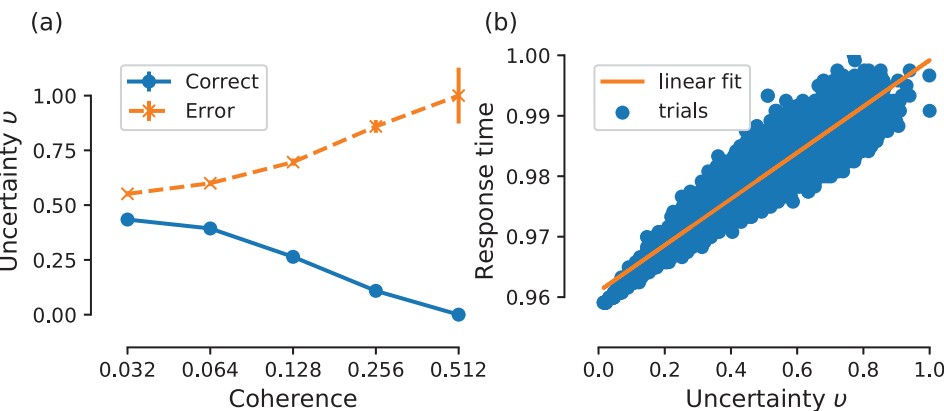

**Fig 7. Simplified model accounts for signatures of decision uncertainty.** (a) Uncertainty as a function of coherence level. '<' pattern: Uncertainty increases (decreases) with coherence level for error (correct) choices. Error bars indicate binomial proportion standard error of mean. (b) Response time as a function of uncertainty. Data points are collected from 36,000 trials. Uncertainty and response time correlate strongly (Pearson's r = 0.95). Trials with zero uncertainty have been discarded.

and coherence in both non-change-of-mind and change-of-mind trials. Specifically, the model's response times for change-of-mind trials were prolonged (compared to non-change-of-mind trials), and did not vary between correct and error trials (Fig 8c).

Importantly, our neural circuit model accounts for the observed positive relationship between response time and changes-of-mind (Fig 8d, cf. Fig 3d). The model suggests that changes-of-mind occur exclusively in trials with long response times. In the experimental data, changes-of-mind could occur even in the trials with fast initial response, but were most probable when response times were longest. The qualitative match between the model and the data suggests that, first, the mechanism underlying changes-of-mind in our experiment and model could be similar, and, second, that these mechanisms are different from other mechanisms that rely solely on post-decision evidence accumulation [7, 9] (see S1 Appendix for analysis of a post-decision evidence accumulation model in the context of the present paradigm).

**Model's high uncertainty is associated with changes-of-mind.** To clarify the link between uncertainty and changes-of-mind in our model, we investigate the pairwise relationships among decision uncertainty, changes-of-mind, and response times (Fig 9).

First, we analysed the mean decision uncertainty level (see Methods) as a function of task difficulty (i.e. coherence level) separately for change-of-mind and non-change-of-mind trials (Fig 9a). As previously mentioned, on average, uncertainty levels are higher in the case of change-of-mind trials compared to non-change-of-mind trials (Fig 6, middle panel), regardless of the outcome of the trial (i.e. correct or error). During such trials, we observed longer initial response times (Fig 8d), which allowed the uncertainty-monitoring population more time to integrate the summed input from the sensorimotor populations, leading to increased greater neural activity (uncertainty levels), and larger total excitatory feedback to the sensorimotor populations.

Interestingly, when sorting the simulated trials based on tertiles of decision uncertainty levels; the probability of a change-of-mind was shown to be highly dependent on the decision uncertainty level (Fig 9b). Hence, our model suggests that changes-of-mind are strongly associated with high decision uncertainty.

To further explain the effect of uncertainty on the dynamics of change-of-mind behaviour in our model, we performed a systematic stability analysis of the value of the population-

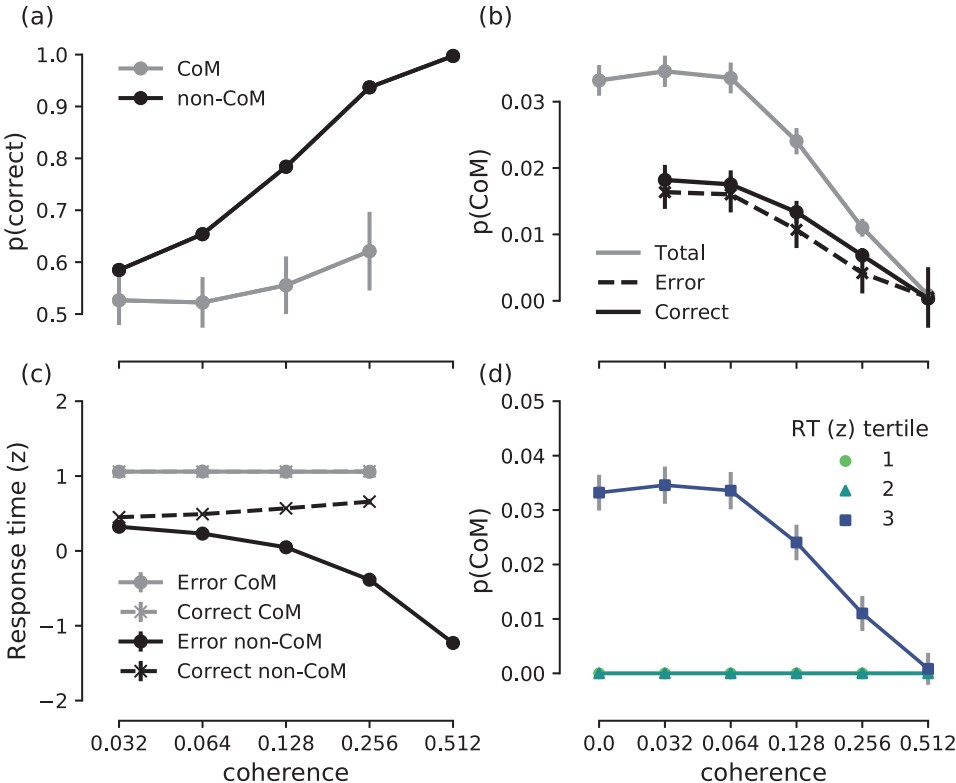

**Fig 8. Model simulation results. (a)** Psychometric function showing choice accuracy as a function of coherence level in the presence (grey) and absence (black) of a change-of-mind. Similarly to the experimental data (Fig 3a), accuracy increases as a function of coherence level, but is lower in change-of-mind trials (grey) compared to non-change-of-mind trials (black). **(b)** Probability of a change-of-mind in all/correct/error trials as a function of coherence. Similar to Fig 3b, probability of a change-of-mind is the highest at low-to-intermediate coherence levels, and decreases sharply at high coherence levels. **(c)** Response times (z-scored) for correct and error change-of-mind and non-change-of-mind trials. The data for correct and error change-of-mind trials overlap, whereas response times for non-change-of-mind trials follow the experimentally observed '<' pattern (Fig 3c). **(d)** Probability of a change-of-mind as a function of coherence level grouped by the tertile of the initial response time. All change-of-mind trials occurred when response times were longest. In all panels, error bars indicate standard error of mean (in panel (c) the error bars overlap with markers).

averaged synaptic gating variable $S_L$ (corresponding to the left-selective sensorimotor population, from Eq 5) with respect to magnitude of uncertainty feedback (Fig 10)—for simplicity, we set the coherence to be zero for this analysis (see [30]).

During the time-course of a change-of-mind trial, the activity of the uncertainty monitoring population gradually increases from zero (Fig 6, red traces). Early in the trial, uncertainty level is close to zero. This leads to low uncertainty excitatory feedback (Fig 10, blue dashed line), and therefore, the network maintains a "winner-take-all" regime with two stable steady states corresponding to two decision states (i.e. Left or Right). As the uncertainty excitatory feedback increases later in the trial, a single stable steady state appears, which corresponds to indecision (Fig 10, region around the red dashed line). However, since the uncertainty feedback is only transient (Fig 6), the network eventually returns to the initial configuration with two stable steady states (Fig 10, blue dashed line). It should be noted that recurrent network reverberation [30] allows neural integration even in the absence of a stimulus. Additionally, due to the stochastic nature of the sensory integration, in cases where the choice ends up being different

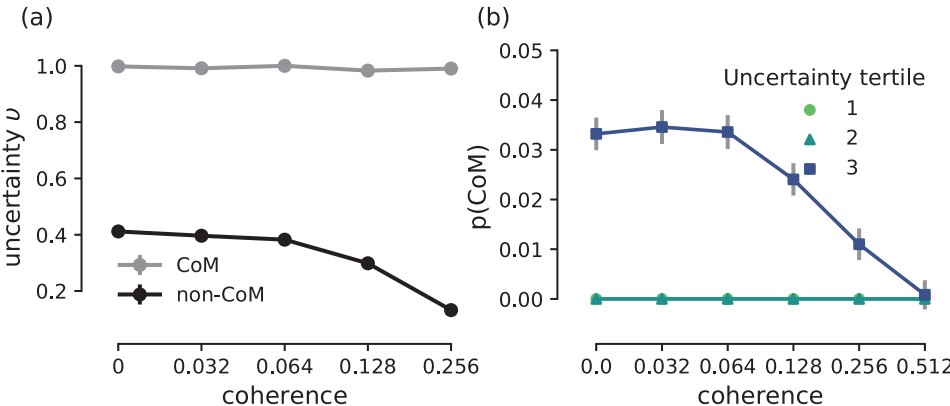

**Fig 9. Model uncertainty is strongly associated with changes-of-mind. (a)** Uncertainty as a function of coherence level split by the type of trial (i.e. change-of-mind vs. non-change-of-mind). Change-of-mind trials are associated with higher uncertainty levels compared to non-change-of-mind trials regardless of the coherence level (see Fig 8c, where response times are predicted to be the same for change-of-mind trials regardless of the coherence level). **(b)** Probability of a change-of-mind as a function of coherence level split by the magnitude of uncertainty level (three tertiles). Changes-of-mind occur only in the highest uncertainty tertile. See Materials and methods for uncertainty level quantification. In both panels, error bars indicate standard error of mean (in panel (a) the error bars overlap with markers).

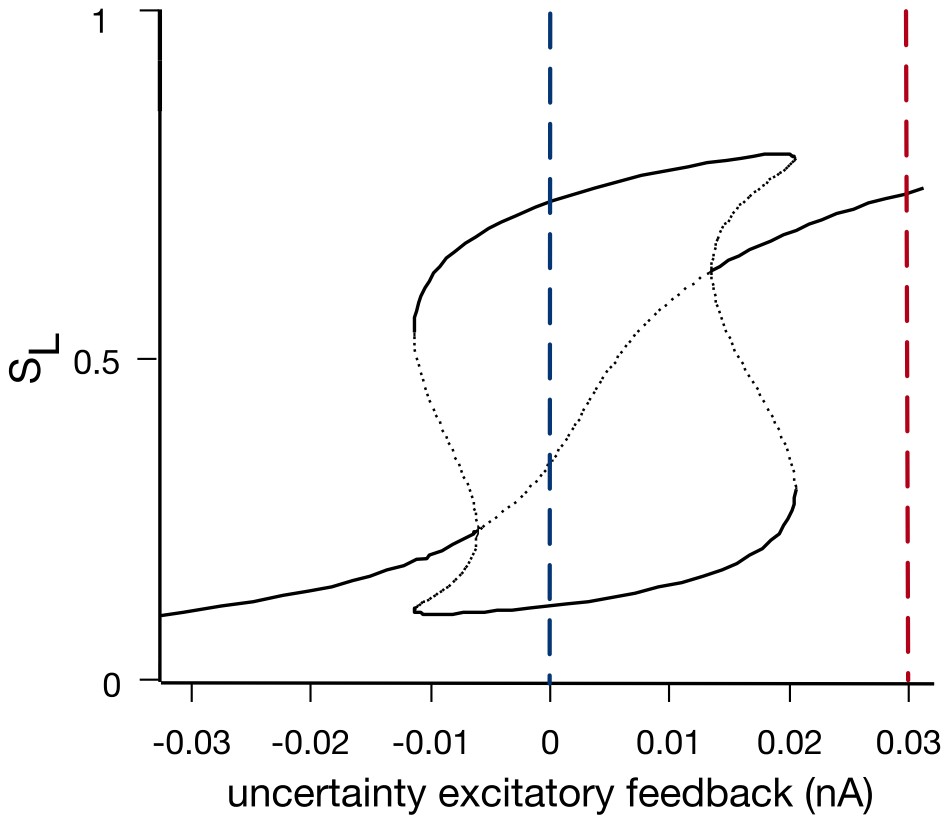

**Fig 10. Bifurcation diagram of the value of the synaptic gating variable $S_L$ (corresponding to the left-selective sensorimotor population) with respect to the magnitude of uncertainty excitatory feedback (at zero coherence level).** No or low uncertainty feedback yields two stable steady states (black solid lines) and one unstable steady state (black dotted lines). This forms a "winner-take-all" regime (blue dashed line). In contrast, high uncertainty feedback (around 0.03) yields only one stable steady state (red dashed line).

from the initial choice (i.e. an initially "losing" neural population "wins" later in the trial), a change-of-mind occurs (see [28] for detailed discussion).

Overall, our model simulations were consistent with our experimental findings; the model predicts that the observed relationship between changes-of-mind and long initial response times is due to high levels of decision uncertainty. Taken together, these results suggest a tight relationship between decision uncertainty and changes-of-mind.

## Discussion

Even well-prepared and thoughtful decisions are occasionally accompanied by a change-of-mind that reverses the initial choice. While considered a hallmark of cognitive flexibility, the mechanism underlying change-of-mind behaviour remains elusive. Here we demonstrate that changes-of-mind can occur even in the absence of additional evidence after the initial decision. These changes-of-mind are associated with slow initial decisions. This is in contrast to previous theories suggesting that changes-of-mind result primarily from post-decision sensory evidence accumulation. The neural circuit model we proposed captures these properties of changes-of-mind, and provides insights into the dynamics of this behaviour, predicting that changes-of-mind in the absence of new post-decision evidence are associated with high levels of decision uncertainty.

Early modelling and experimental work on error correction [4, 5, 13, 14] suggested that error correction can be characterised by an evidence accumulation process in which initial error responses are reversed (or corrected) by new incoming evidence that negates the initial erroneous judgement. Recent experimental investigations of changes-of-mind build on the same framework, reinforcing the link between error correction and changes-of-mind. These studies focus on changes-of-mind that are primarily driven by the noisy accumulation of additional sensory evidence, which either arrives late due to processing delays [7, 9, 12], or is provided separately after the initial decision [6, 10]. In contrast, majority of changes-of-mind in our study occurred later than 450ms after the stimulus offset, which is outside the hypothesised delayed information processing window [26]. This supports the notion that the changes-of-mind we observed are not associated with post-decision accumulation of delayed evidence or explicitly provided new information, as opposed to previous studies [4, 7, 9, 10, 12, 14].

Despite the absence of new post-decision evidence, changes-of-mind in our experiment improved the initial decisions in the trials with intermediate-to-high stimulus coherence (Fig 3a), which was also the case in previous studies [7, 9, 27]. However, in sharp contrast to these studies, we found a positive relationship between initial response times and subsequent changes-of-mind: changes-of-mind were most likely to occur in trials with prolonged response times (Fig 3d). Hypothetically, this relationship can arise when initial response times and the frequency of changes-of-mind are both associated with high decision uncertainty. Future experimental work can directly test this hypothesis by requiring participants to report their confidence retrospectively [10] or in parallel with their choice [9, 31] using the fixed stimulus viewing duration paradigm employed in this study. In the absence of such confidence reports in our paradigm, here we take a complementary approach—we demonstrate that our mechanistic model of decision uncertainty and changes-of-mind could account for our experimental findings.

In our mechanistic model, decision uncertainty is continuously monitored by a top-down uncertainty-monitoring neuronal population. Importantly, through an excitatory feedback loop, decision uncertainty continuously affects the neuronal integration dynamics, which occasionally triggers a change-of-mind in trials with high uncertainty. Our model provides insights into the mechanism of changes-of-mind observed in our experiment. Unlike

extensions of the drift-diffusion model [7, 15, 16, 25] and previous attractor network models [32], our model does not rely solely on post-decisional sensory evidence accumulation to induce changes-of-mind. Importantly, the mentioned models were not designed to account for the situations where no additional post-decision evidence is available after the initial decision. In contrast, in our model, the final outcome of a trial is dynamically affected by decision uncertainty monitored during the stimulus presentation. In trials with high levels of uncertainty, strong excitatory feedback from the uncertainty-monitoring neuronal population leads to a delayed initial response, due to the high competition between sensorimotor populations (Fig 6, [28]). At the same time, this increased uncertainty could lead to a change-of-mind (Fig 9, [28]). Through this mechanism, our neural circuit model accounted for the observed positive relationship between initial response time and changes-of-mind (Fig 8c and 8d). This finding was further supported by the analysis of the probability of a change-of-mind as a function of uncertainty level (Fig 9b). More specifically, our model predicted that in situations where no new stimulus-related evidence could be sampled after the initial decision, changes-of-mind are most likely to occur during trials with high levels of uncertainty. Future work could test this by providing neural recordings of brain regions that encode decision uncertainty [19, 22] during changes-of-mind in a fixed-duration perceptual discrimination task.

The model proposed here is a variant of the previously developed model [28], which was inspired by neurophysiological recordings from brain regions encoding decision confidence or, reciprocally, decision uncertainty [19, 22]. In this work, we have reduced the uncertainty-monitoring module of the original model to lay bare its essential functions. In addition to the similarity of the neural profile of our model's uncertainty-encoding population (Fig 6) to existing recordings [19, 22], this reduced model accounts for some of the main characteristics of decision uncertainty [22, 29, 31] (Fig 7). It should be noted that this reduced implementation of decision uncertainty could arguably be less neurobiologically plausible compared to the previous model, as the latter involves a canonical cortical column structure—inhibitory-excitatory pair of neural populations [33, 34]. Importantly, we have not changed how the other (i.e. sensorimotor and motor) modules are described, which can serve as further validation of previous modelling work [28]. In particular, the sensorimotor module is based on a previously developed mean-field model with biologically derived variables [30]. Therefore, the model can be tested in future studies using neurophysiological recordings within similar experimental task paradigms to allow the distinguishing between different models [35]. It should also be noted that such reductions of biophysical realisations of accumulate-to-bound models [36] can be linked back to simpler models of decision making [15, 30, 37–39].

Decision uncertainty or confidence is closely related to (decision) conflict monitoring. Computational models of cognitive control have used conflict monitoring to account for various behavioural aspects of decision uncertainty [40]. For instance, such models have been shown to account for post-error slowing, attentional bias, error-related negativity, error-prediction, and neuromodulatory processing [41–45]. In conflict monitoring models, conflict is usually modelled by a dedicated conflict monitoring unit that takes as an input the instantaneous activities of the competing (e.g. decision) units, with the output being the multiplication of these activation rates. This representation accounts for various signatures of conflict. Specifically, conflict typically increases with increasing activities of the competing units, and conflict level is highest when competing units reach their maximal activities. Unlike these conflict-monitoring models, which are mainly based on abstract sigmoidal- or logistic-like activation (input-output) functions, the original decision uncertainty module [28] is inspired by a canonical cortical microcircuit model [33, 34], which the current model is reduced from. Further, its input-output function is represented by a simple threshold-linear model, i.e. neural activity saturation is unnecessary. Importantly, the quantification of decision uncertainty involves the more biologically plausible

mechanism of summation of inputs from presynaptic neurons [46], as compared to the multiplication of inputs in conflict-monitoring models. Moreover, our model mechanistically relates the decision uncertainty to continuous motor output dynamics.

Overall, our work provides a computational framework that explains changes-of-mind in the absence of new post-decision evidence, from sensory integration to uncertainty monitoring and motor output, by mechanistically linking decision uncertainty and changes-of-mind. Taken together, our findings highlight the role of top-down metacognitive processes in changes-of-mind.

## Methods and materials

### Ethics statement

The study protocol was approved by NUI Galway Research Ethics Committee, REF: 15/DEC/ 02. Written consent was obtained.

### Participants

Thirteen healthy adults (four male, nine female, 20 to 44 years old) were recruited to participate in the experiment in exchange for a €30 gift voucher. The data were collected in two locations: four participants performed the task in Galway (Ireland); three years later, nine participants performed the same task in Dresden (Germany). We tested all results for robustness by analyzing the data from these two groups separately. Since all the results hold for both datasets, we analyze the two datasets together in this paper. Data from two participants were excluded due to atypical proportion of changes-of-mind (9% and 13%), resulting in $N = 11$ participants whose data were analyzed further. Ten participants were right-handed, one was left-handed; all participants had normal or corrected-to-normal vision.

### Apparatus

Participants performed the task in a sitting position in front of a desktop computer equipped with a 24 inch monitor (1920 by 1080 pixels). Mouse cursor coordinates were sampled at 60 Hz during stimulus presentation and at 100 Hz on the response screen. The mouse cursor speed was set to 50% in the Windows 7 mouse properties settings; the "Enhance pointer precision" option was disabled. In four participants, eye movements were recorded using an eye-tracker, but were not analyzed. The stimulus presentation software was programmed in Python using PsychoPy [47] and PyGaze [48].

### Task

Participants performed a perceptual decision-making task (Fig 1). Each trial started when a participant clicked the start button located at the bottom of the screen. After a random delay (uniformly distributed over 700–1000ms), the RDK was presented for fixed duration of 800 ms, followed by a screen with two response options. A participant then moved the mouse cursor from the bottom of the screen to one of the top corners and then clicked on a response area to indicate their choice. Immediately after that, the feedback (green circle for correct responses, red for incorrect) was presented for 300 ms, followed by a fixation cross for another 300 ms. Participants were instructed to respond as fast and accurately as possible.

### Stimuli and procedure

The random dot kinematogram (RDK) algorithm [23, 24] was used for stimulus presentation. The dots were presented in a 5˚ square aperture (distance between the monitor and the

participant's eyes was approximately 80cm). During each frame, 3 dots were displayed. The monitor used for stimulus presentation has a refresh rate of 60 Hz. This entails that the resulting dot density is 16.7 dots per $deg^2$ per sec. The dot velocity was set to 5˚/sec. On each trial, the direction of the stimulus (left or right) was determined randomly. The probability for each dot to move coherently on a given frame was determined by the coherence parameter.

The experiment consisted of four sessions held on four different days over the span of 4 to 18 days. Each session included 600 trials, grouped into ten blocks of 60 trials. Each block contained 10 trials for each of the six coherence levels (0, 0.032, 0.064, 0.128, 0.256, 0.512), randomly shuffled. In total, each participant completed 2400 trials, 400 for each coherence level.

## Data analysis

A trial was labelled as a change-of-mind if a response trajectory deviated from the (implicit) vertical centre line towards the unchosen option by more than 100 pixels (in the *x*-direction). However, deviations which could have resulted from erratic movements in the early stages of response were ignored: if the threshold of 100 pixels in the horizontal direction was crossed in the bottom 10% of the response area, the trial was not labelled a change-of-mind. After 120 trials with more than one changes-of-mind were excluded from all analyses, this resulted in 775 change-of-mind trials in total.

Response time was measured as the time between the stimulus offset and the response onset. Response onset was determined as the onset of the first hand movement resulting in a mouse cursor displacement greater than 100 pixels; therefore, small movements resulting, e.g., from hand tremor did not affect RT measurement. In trials where participants initiated the response before the stimulus offset, the response time was considered to be negative. Overall, 30% of all trials had negative response time, with 52% of all negative response times observed at two highest coherence levels.

For mixed-effects statistical models (Tables 2 and 3), R package lme4 was used. In all models, random effects of participant were included to account for individual differences, with the maximum random effects structure permitting model convergence. For testing the hypothesis that changes-of-mind improve accuracy, the R implementation of the exact binomial test (binom.test) was used.

## A reduced neural circuit model of uncertainty

We used a simplified version of our previous neural circuit model of decision uncertainty and change-of-mind [28], in which the dynamics of uncertainty-encoding is described using one neural population (i.e. dynamical variable). The modelling of the sensorimotor and motor (i.e. hand) populations was unchanged (see below). The dynamics of the uncertainty-encoding neural population:

$$\tau_{mc} \frac{dy_{HU}}{dt} = [J_{VHU}(H_L + H_R) - g]_+ - y_{HU} \qquad (1)$$

where $[\;]_+$ denotes a threshold-linear input-output function. Synaptic coupling constant between the uncertainty-encoding population and the sensorimotor neural populations is denoted by $J_{VHU}$. $H_L$ and $H_R$ denote the neuronal population firing rates of the sensorimotor populations. At the beginning of a trial, some top-down inhibition is activated (g = 1000 nA) and 600 ms after stimulus onset from. Further, g is reactivated (with a value of 3000 nA) when the activity of one of the sensorimotor neural populations reaches a threshold (42.5 Hz). The result is a phasic activity response of the high uncertainty-encoding population that is

reminiscent of recent neural recordings from the prefrontal cortex and medial frontal cortex during error correction post-decisional accumulation [10, 19, 22].

## Uncertainty level quantification

Similar to our previous work [28], we used the maximum firing rate value of the uncertainty-encoding neuronal population as a decision uncertainty measurement. In the case of trial-averaged measurements, we calculated the trial-averaged and SEM of these maximal values for each coherence level. We then normalised these values using min-max normalisation. This normalisation can be described by:

$$X' = \frac{X - X_{min}}{X_{max} - X_{min}} \tag{2}$$

## Classifying model outputs and changes-of-mind

Response time is recorded in the model as the moment the activity of one the sensorimotor populations reaches 42.5 Hz. In the simulation of the hand neuronal population, the target is fixed at 42.5 Hz. A simulated trial is classified as a change-of-mind if a reversal of dominance in firing rates between the two hand neuronal population occurs. A threshold of 2 Hz was used for the absolute difference in magnitude.

## Mapping the activity of the hand neuronal populations onto the X positional space

To reproduce the typical trial dynamics observed in our experiment, we used a simple linear function to approximate the hand X position as a function of the neuronal firing rate [28]. This approximation can be described as follows:

$$x = q(y_{LH} - y_{RH}) \tag{3}$$

where $q$ is some scaling factor. This scaling factor is determined as follows:

$$q = |C_{pos}|/H_{th} \tag{4}$$

where $C_{pos}$ denotes the X position of the choice target (760px). $H_{th}$ is the hand target threshold (17.4Hz).

## Modelling the sensorimotor populations

We used a reduced spiking neural network model [49] described by two NMDA-mediated synaptic gating variables (i.e. dynamical variables) [30]. These two variables can be described by:

$$\frac{dS_L}{dt} = -\frac{S_L}{\tau_s} + (1 - S_L)\gamma H_L(x_L , x_R) \tag{5}$$

$$\frac{dS_R}{dt} = -\frac{S_R}{\tau_s} + (1 - S_R)\gamma H_R(x_R , x_L) \tag{6}$$

where:
$\gamma$: A constant
$\tau_S$: The synaptic gating time constant.
$H$: A nonlinear input-output function (see below).

The firing rates of sensorimotor neuronal populations can be described by:

$$H_i = G_S\left(\frac{ax_i - b}{1 - e^{-d(ax_i - b)}}\right) \tag{7}$$

$$x_i = J_{N,ii}S_i - J_{N,ij}S_j + I_0 + I_i + J_{mc0}y_{HU} \tag{8}$$

$$I_i = J_{A,ext}\,\mu_0\left(1 \pm \frac{\varepsilon}{100\%}\right) \tag{9}$$

where: $a$, $b$, $d$: Parameters for the input-output function fitted to a leaky integrate-and-fire neuronal model [49].
$I_0$: A constant denoting effective input bias.
$c$: Coherence level for a given trial.
$J_{N, ii}$ and $J_{N, ij}$: Synaptic connections strength.
$I_i$: Effective stimulus input to population $i$.
$\mu_0$: The stimulus strength constant.
$J_{A, ext}$: External synaptic coupling strength.
$J_{mc0}$: The strength of the excitatory feedback from the uncertainty-encoding population.
$G_S$: Input-output function gain.

## Modelling the hand populations

We used a threshold linear function to model the action outputs via hand. We achieved persistent neural activity for the hand populations via mutual inhibition to create a line attractor model [50]. The dynamics of the two hand neuronal populations can be described by:

$$\tau_h \frac{dy_L}{dt} = [H_L - J_{N,LR}\, y_{HR} - g]_+ - y_{HL} \tag{10}$$

$$\tau_h \frac{dy_R}{dt} = [H_R - J_{N,LR}\, y_{HL} - g]_+ - y_{HR} \tag{11}$$

where:
$[\ ]_+$: Threshold-linear input-output function.
$H_L$ and $H_R$: Firing rates of the sensorimotor populations (see above).
$J_{N, LR}$: Synaptic coupling strength between left and right neuronal populations (i.e. effectively inhibitory connection).
$g$: Top-down inhibition that is deactivated when the neural activity of a sensorimotor population reaches the response threshold.

## Model simulation and analysis

The code to simulate the model (and analyse its outputs) was written in MATLAB. The code was tested against one version of MATLAB (2018a, on a Mac OS X workstation). The model parameters are summarised in Table 4. The model was simulated for 6000 trials per condition, using the same task specifications outlines above (i.e. 800ms fixed-duration stimulus). 3.4% of the trials were non-decision trials (i.e. target threshold was not reached) and were discarded. We used XPPAUT [51] for phase-plane analysis and parameter search. For within trial dynamics, we used a forward Euler-Maruyama numerical integration scheme. Integration time step set to 0.5ms. Smaller time steps did not affect our results.

**Table 4. Table of model parameter values.** Parameters $\tau_S$, $a$, $b$, $d$, $I_0$, $\mu_0$, $J_{A,ext}$ were directly adapted from [30]. Parameters $\tau_h$, $t_{mc}$, $J_{mc0}$, $J_{N,\,ij}$, $H_{th}$ were directly adapted from [28], and parameters $J_{VHU}$, $J_{N,\,LR}$, $J_{N,\,RL}$, $G_S$, $S_{th}$, $J_{N,\,ii}$ were tuned to account for the behavioural data.

| Parameter | Description | Value |
|---:|---:|---:|
| $\tau_S$ | Sensorimotor time constant | 100ms |
| $\tau_h$ | Motor time constant | 50ms |
| $t_{mc}$ | Uncertainty population time constant | 150 ms |
| $a$ | Input-output function parameter | 270 (V nC)$^{-1}$ |
| $b$ | Input-output function parameter | 108 Hz |
| $d$ | Input-output function parameter | 0.154 s |
| $I_0$ | External tonic input | 0.3255 nA |
| $J_{N,\,ii}$ | Self-excitation strength | 0.248 nA |
| $J_{N,\,ij}$ | Inhibition strength (sensorimotor) | 0.0497 nA |
| $\mu_0$ | Baseline stimulus input | 30 Hz |
| $J_{A,\,ext}$ | External input synaptic strength | 0.00052 nA Hz$^{-1}$ |
| $J_{mc0}$ | Uncertainty feedback strength | 0.002 |
| $J_{VHU}$ | External input strength (uncertainty) | 10 nA |
| $J_{N,\,LR}$ | Inhibition strength (motor) | 1 nA |
| $J_{N,\,RL}$ | Inhibition strength (motor) | 1 nA |
| $S_{th}$ | Sensorimotor module threshold | 42.5 Hz |
| $H_{th}$ | Hand module threshold | 17.4 Hz |
| $G_S$ | Sensorimotor input-output gain | 1.12 Hz |

## Code and data availability

All code and data needed to evaluate or reproduce the figures and analysis described in the paper are available online at: https://osf.io/y385t/. This includes the collected data, and the code for stimulus presentation, model simulation, and data analysis.

## Supporting information

**S1 Appendix. Analysis of the extended drift-diffusion model.**
(PDF)

## Author Contributions

**Conceptualization:** Nadim A. A. Atiya, Arkady Zgonnikov, Denis O'Hora, KongFatt Wong-Lin.

**Data curation:** Nadim A. A. Atiya, Arkady Zgonnikov, Denis O'Hora, Martin Schoemann, Stefan Scherbaum.

**Formal analysis:** Nadim A. A. Atiya, Arkady Zgonnikov.

**Funding acquisition:** Nadim A. A. Atiya, Arkady Zgonnikov, Denis O'Hora, Stefan Scherbaum, KongFatt Wong-Lin.

**Investigation:** Nadim A. A. Atiya, Arkady Zgonnikov, Martin Schoemann.

**Methodology:** Nadim A. A. Atiya, Arkady Zgonnikov, Denis O'Hora, KongFatt Wong-Lin.

**Project administration:** Denis O'Hora, KongFatt Wong-Lin.

**Resources:** Denis O'Hora, Stefan Scherbaum, KongFatt Wong-Lin.

**Software:** Nadim A. A. Atiya, Arkady Zgonnikov.

**Supervision:** Denis O'Hora, Stefan Scherbaum, KongFatt Wong-Lin.

**Validation:** Nadim A. A. Atiya, Arkady Zgonnikov.

**Visualization:** Nadim A. A. Atiya, Arkady Zgonnikov.

**Writing – original draft:** Nadim A. A. Atiya, Arkady Zgonnikov.

**Writing – review & editing:** Nadim A. A. Atiya, Arkady Zgonnikov, Denis O'Hora, Martin Schoemann, Stefan Scherbaum, KongFatt Wong-Lin.

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
