## [Decision Letter · Decision Letter 0]

27 Jul 2019

Dear Dr Wong-Lin,

Thank you very much for submitting your manuscript 'Changes-of-mind in the absence of new post-decision evidence' for review by PLOS Computational Biology. Your manuscript has been fully evaluated by the PLOS Computational Biology editorial team and in this case also by independent peer reviewers. The reviewers appreciated the attention to an important problem, but raised some substantial concerns about the manuscript as it currently stands. While your manuscript cannot be accepted in its present form, we are willing to consider a revised version in which the issues raised by the reviewers have been adequately addressed. We cannot, of course, promise publication at that time.

Sincerely,

Samuel J. Gershman

Deputy Editor

PLOS Computational Biology

[LINK]

Reviewer's Responses to Questions

**Comments to the Authors:**

Reviewer #1: In this very interesting paper, Atiya et al analyze the changes of mind (CoM) that occur in the absence of new post-decision information. Previous papers on CoM have invariably used paradigms where new evidence arrives after the decision has already been made. Therefore, the CoMs observed in those studies are easily interpreted as simply the newest evidence being stronger than the older one, which makes it a much less interesting phenomenon. The authors here make a significant advance by showing that CoMs occur even in the absence of new evidence. However, critically, the signatures associated with these CoMs are very different than the signatures in previous studies with different designs. These are important discoveries. I only have major suggestions regarding the presentations of the results.

My only major comment is that, as it stands, the paper is rather difficult (and sometimes even frustrating to read). There are a number of reasons. Several sections start with a number of assertions in the first paragraph that are only later supported by stats in the second paragraph. I’d suggest introducing the stats the first time a claim is made; otherwise, the reader has to keep in working memory all claims that were previously made and check whether there’s support for them only later. The Results section starts with a heading titled “Cognitive task and neural circuit model”. That section includes some background and a single “teaser” result followed by introducing the model when it was not yet clear what needs explaining (the 3% CoM alone could certainly be explained in many other ways).

More generally, the organization of the manuscript makes it seem (at least for a while) that it’s mostly about the authors’ model. However, I think that the strongest part of this paper is the behavioral results, which become somewhat secondary in the current writeup. I’d suggest changing the organization where the behavioral effects are presented first, followed by the eDDM results, and only then followed by the authors’ model. This would also highlight that the model can fit patterns of results that are not “easy” to fit as eDDM cannot fit them.

Minutiae

1. It was rather frustrating to read this paper as the figures were placed at the end (journals should not require this and if they do, authors should ignore it for the first submission in the interest of everybody) and figure legends appeared before the figures were references in the main text. I assume that none of this is the authors’ choice but I suggest re-formatting future submissions.

2. P.5 “However, in change-of-mind trials, choice accuracy was on average lower than in non-change-of-mind trials.” This statement appears to me to be completely unrelated to the one above (why should the trends in these two analyses be related?) but the use of the word “however” suggests otherwise.

3. P.5 “However, although there was an ascending trend for the rate of correct changes-of-mind with increasing coherence level, the trend was not as steep as that observed in the experiment.” Some sort of statistical analysis would help here.

4. P.5 “followed the ’<’ pattern” – may be clearer to spell this out.

5. How are the 294 CoMs distributed among the 4 subjects?

6. I may have missed this but how was a CoM defined exactly? What is the cutoff for determining that subjects moved sufficiently towards one target before changing? More speculatively, given that the authors have continuous measurements, can CoM itself also be defined as a graded variable (I’m not suggesting that the authors do this but it may be worth discussing the possibility)?

7. I commend the authors for sharing their data and codes. They write that “All codes and data needed to evaluate or reproduce the figures and analysis described in the paper are available online at: https://osf.io/y385t/. This includes the collected data, and the code for stimulus presentation, model simulation, and data analysis.” However, when I followed the link, I didn’t find any codes – it appeared that what is there is only raw(-ish) data. I also suggest including a file that provides a key to what is contained in each folder as well as a license document (typically CC-BY is a good idea).

Reviewer #2: The manuscript examines the phenomenon of change of mind (COM) in perceptual decisions with moving dots stimuli, and with responses that are required after stimulus offset. This differs from some previous studies that examined COM responses in a situation where response terminate the stimulus. The manuscript reports data from 4 participants, which shows about 3% COM responses, and which appear to increase with stimulus difficulty (low coherence) and with RT (more COM at slow responses). These two pattern are different from what was reported in conditions where the response terminates the stimulus, and the authors propose a computational model based on uncertainty monitoring (rather than integration of extra-evidence), which accounts for qualitative aspects of the data.

While I find the question – do COM require extra-evidence and if not what mechanism can account for them – interesting, and the work carried out of good professional standard, there are three important reasons why I do not feel that, this paper is ready for publication in this Journal. First, the experiment includes only 4 participants. Second the literature discussed in quite incomplete, and finally, the paper is difficult to read and at many places the Method and the Computational mechanism proposed is not clear (and appears somewhat adhoc). Below I elaborate on these 3 points before I provide a number of more specific comments, which the authors should address in a potential revision.

1. participants

4-participants is really below the standard in cognitive psychology (a set of 8-12 would be a minimum requirement). While psycho-physical studies do sometimes uses such small designs, this is usually done when each subject is tested over repeated sessions, and individual data are presented (showing the same pattern in each participant). This is definitely not the case here, especially as the critical COM trials are quite rare (<3%).

2. Literature.

There is much literature, both experimental and computational, which seems quite related to the issues discussed here, which is neither mentioned nor discussed. On the experimental side there are numerous papers by Rabbitt P. and colleagues, on error-correction (or double responding), which seems to me to probe a quite similar phenomenon to the one discussed in this paper (see also study by Bronfman et al., 2015, Exp-2, which reports more correct than incorrect COM). Even if those did not use moving dots stimuli, it will help to discuss differences (or commonalities) in the experimental results and in the mechanisms that were proposed. Second, there are computational models that extended the drift-diffusion model framework to competitive interactions (e.g., Bogacz et al. 2006), and to conflict monitoring (Botvinick et al., 2001). The latter, in particular, is somewhat similar to the uncertainty mechanism proposed here (except that it is multiplicative rather than additive). Since conflict and uncertainty seem related, a discussion of differences would be much needed (for example are there important differences between multiplicative and additive mechanisms, and which is more plausible?) A list of relevant references is provided below.

3. Computational Model

I found it quite difficult to understand the model, which could be partly due to the very compressed writing in the main text. As I don’t think that this journal has a serious limitation on the number of words, I would propose to the authors to provide systematic and clear descriptions of the model, in the main text, rather than asking the readers to work hard by shifting back and forth to the Method or Supplement sections. More important, it appears to me that many of the modeling assumptions are somewhat adhoc. Just to give one example, while the sensorymotor populations are described by a “reduced spiking model” (Eq. 5-9), the hand-populations (eq. 10-11) and the uncertainty population are described by more simple firing-rate equations. As the present paper does not address any neural data, I find this variability unmotivated and somewhat distracting. Thus I would suggest to the authors to rely on the same dynamics throughout the paper (I would prefer the simpler ones; 10-11), unless there is an important reason why this cannot be done. Also, there are at present 13 model parameters and it is not very clear why they are picked as they are (was there any data fitting?)

Finally, while the simulations show that the model (more or less) accounts for the phenomena (but I also see some clear deviations in Fig 3), the paper does not provide a transparent explanation why it does so. How does the uncertainty estimation drive (slow-RT) changes of mind? Lets say the subject experiences uncertainty at time t, why/how does this trigger a change of mind? I suppose it has to do with the input from the uncertainty population to the sensorymotor populations, but why should this reverse the earlier choice rather than amplify it?) Also how does the uncertainty relate to the stimulus coherence (is higher moving-dot coherence generate more or less uncertainty, and why? How does the coherence affect the input into the sensorymotor populations). Showing that the model accounts for the data is good, but it would be even better to provide an explanation why it does so. Also I would suggest, that the authors present in the Introduction, the challenge that the present model needs to clear out (the slow COM in the absence of extra-evidence), so that the readers are more focused on this central aim.

Specific comments

p. 3. Why is the 1-population implementation less plausible?

p. 3 Bottom: eliminate “more” in “more simplified”; change “could still replicate” to “replicates”, etc.

p. 4. Explain in main text how the COM is identified in the data

Fig. 1 (Model) is not very effective. The sensory and hand modelus should be arranged vertically with arrows from one to the other.

Fig. 5. Is also not very clear. Is this a simulation or an experimental data pattern? If its a simulation one needs more details.

References

Bogacz R1, Brown E, Moehlis J, Holmes P, Cohen JD. (2006). The physics of optimal decision making: a formal analysis of models of performance in two-alternative forced-choice tasks.

Psychol Rev., 113(4):700-65.

Botvinick MM1, Braver TS, Barch DM, Carter CS, Cohen JD. (2001). Conflict monitoring and cognitive control. Psychol Rev, 108: 624-52.

Bronfman ZZ, Brezis N, Moran R, Tsetsos K, Donner T, Usher M. (2015). Decisions reduce sensitivity to subsequent information. Proc. R. Soc. B.

Rabbitt, P. (1966). Errors and error correction in choice–response tasks. Journal of Experimental Psychology, 71 , 264–272.

Rabbitt, P. (1967). Time to detect errors as a function of factors affecting choice-response time. Acta psychologica, 27 , 131–142.

Rabbitt, P. (1968). Three kinds of error-signalling responses in a serial choice task. Quarterly Journal of Experimental Psychology, 20 , 179–188.

Rabbitt, P. (1969). Psychological refractory delay and response-stimulus interval duration in serial, choice-response tasks. Acta Psychologica, 30 , 195–219.

Rabbitt, P. (2002). Consciousness is slower than you think. The Quarterly Journal of Experimental Psychology Section A, 55 , 1081–1092.

Rabbitt, P., Cumming, G., & Vyas, S. (1978). Some errors of perceptual analysis in visual search can be detected and corrected. The Quarterly Journal of Experimental Psychology, 30 , 319–332.

Rabbitt, P., & Rodgers, B. (1977). What does a man do after he makes an error? An analysis of response programming. Quarterly Journal of Experimental Psychology, 29 , 727–743.

Rabbitt, P., & Vyas, S. (1981). Processing a display even after you make a response to it. How perceptual errors can be corrected. The Quarterly Journal of Experimental Psychology Section A, 33 , 223–239.

**Have all data underlying the figures and results presented in the manuscript been provided?**

Reviewer #1: Yes

Reviewer #2: Yes

PLOS authors have the option to publish the peer review history of their article (what does this mean?). If published, this will include your full peer review and any attached files.

Reviewer #1: No

Reviewer #2: No

---

## [Decision Letter · Decision Letter 1]

8 Nov 2019

Dear Dr Wong-Lin,

We are pleased to inform you that your manuscript 'Changes-of-mind in the absence of new post-decision evidence' has been provisionally accepted for publication in PLOS Computational Biology.

In the meantime, please log into Editorial Manager at https://www.editorialmanager.com/pcompbiol/, click the "Update My Information" link at the top of the page, and update your user information to ensure an efficient production and billing process.

One of the goals of PLOS is to make science accessible to educators and the public. PLOS staff issue occasional press releases and make early versions of PLOS Computational Biology articles available to science writers and journalists. PLOS staff also collaborate with Communication and Public Information Offices and would be happy to work with the relevant people at your institution or funding agency. If your institution or funding agency is interested in promoting your findings, please ask them to coordinate their releases with PLOS (contact ploscompbiol@plos.org).

Thank you again for supporting Open Access publishing. We look forward to publishing your paper in PLOS Computational Biology.

Sincerely,

Samuel J. Gershman

Deputy Editor

PLOS Computational Biology

Reviewer's Responses to Questions

**Comments to the Authors:**

Reviewer #1: The authors have addressed all of my comments. I congratulate them on a very interesting paper.

**Have all data underlying the figures and results presented in the manuscript been provided?**

Reviewer #1: Yes

PLOS authors have the option to publish the peer review history of their article (what does this mean?). If published, this will include your full peer review and any attached files.

Reviewer #1: No

---

## [Editor Report · Acceptance letter]

23 Jan 2020

PCOMPBIOL-D-19-00843R1 

Changes-of-mind in the absence of new post-decision evidence

Dear Dr Wong-Lin,

I am pleased to inform you that your manuscript has been formally accepted for publication in PLOS Computational Biology. Your manuscript is now with our production department and you will be notified of the publication date in due course.

With kind regards,

Matt Lyles
